# Methyl Jasmonate Induces Genes Involved in Linalool Accumulation and Increases the Content of Phenolics in Two Iranian Coriander (*Coriandrum sativum* L.) Ecotypes

**DOI:** 10.3390/genes13101717

**Published:** 2022-09-24

**Authors:** Farzad Kianersi, Davood Amin Azarm, Farzaneh Fatemi, Alireza Pour-Aboughadareh, Peter Poczai

**Affiliations:** 1School of Environmental Sciences, University of Guelph, 50 Stone Road East, Guelph, ON N1G 2W1, Canada; 2Department of Horticulture Crop Research, Isfahan Agricultural and Natural Resources Research and Education Center, AREEO, Isfahan P.O. Box 81785-199, Iran; 3Department of Agronomy and Plant Breeding, Faculty of Agriculture, Bu-Ali Sina University, Hamedan P.O. Box 6517838695, Iran; 4Seed and Plant Improvement Institute, Agricultural Research, Education and Extension Organization (AREEO), Karaj P.O. Box 3158854119, Iran; 5Botany Unit, Finnish Museum of Natural History, University of Helsinki, P.O. Box 7, FI-00014 Helsinki, Finland

**Keywords:** coriander, total flavonoid content, total phenolic, linalool, methyl jasmonate, gene expression

## Abstract

The medicinal herb coriander (*Coriandrum sativum* L.), with a high linalool (LIN) content, is widely recognized for its therapeutic benefits. As a novel report, the goals of this study were to determine how methyl jasmonate (MeJA) affects total phenolic content (TPC), LIN content, flavonoid content (TFC), and changes in gene expression involved in the linalool biosynthesis pathway (*Cs**γTRPS* and *CsLINS*). Our findings showed that, in comparison to the control samples, MeJA treatment substantially enhanced the TPC, LIN, and TFC content in both ecotypes. Additionally, for both Iranian coriander ecotypes, treatment-induced increases in *CsγTRPS* and *CsLINS* expression were connected to LIN accumulation in all treatments. A 24 h treatment with 150 µM MeJA substantially increased the LIN content in the Mashhad and Zanjan ecotypes, which was between 1.48 and 1.69 times greater than that in untreated plants, according to gas chromatography–mass spectrometry (GC-MS) analysis. Our findings demonstrated that MeJA significantly affects the accumulation of LIN, TPC, and TFC in Iranian *C. sativum* treated with MeJA, which is likely the consequence of gene activation from the monoterpene biosynthesis pathway. Our discoveries have improved the understanding of the molecular mechanisms behind LIN synthesis in coriander plants.

## 1. Introduction

In medications or as byproducts for cosmetics, personal care, incense, and nourishment, aromatic and medicinal plants are used to prevent and cure illnesses and to preserve health [1]. The popularity of these plants continues to rise, as people’s interest in natural resources rapidly increases [2,3]. Coriander (*C. sativum* L.), a hardy annual plant in the Apiaceae family, is now grown in a number of temperate regions. Although the exact origin of coriander is unknown, some writers have suggested that it originated in the Middle East, the Mediterranean, and the Near East [4,5,6]. Coriander has been used since antiquity as a spice, an odorant, and in folk medicine [7]. The stem, leaves, and fruits of coriander contain the primary volatile component, linalool, which accounts for 60–70% of the essential oil [8,9]. Geraniol, pinene, limonene, geranyl acetate, terpinene, and borneol are other substances found in coriander [7]. Its essential oil (EO) exhibits biological effects, including anti-inflammatory, analgesic, antibacterial, antifungal, and insecticidal qualities [10,11].

The complex combination of mono- and sesquiterpenes that make up essential oils may also include trace quantities of other metabolites. Many spices and plants present unique characteristics due to the volatility and amount of these molecules [12]. The time of day, physiological state of the plant organ, and environmental factors, including temperature and light intensity, among other factors, all affect the chemical makeup of the essential oils [12]. The plant tissues, including flowers, leaves, stems, buds, seeds, and roots, contain plant terpenoids, also known as isoprenoids. These compounds are composed of five carbon isoprene units. Because many terpenoids (those that make up EOs) are cytotoxic, plants have evolved specific structures for storing these chemicals [13]. Terpenoids are divided into seven primary groups based on the amount of isoprene units that make up their backbone structure [14]. The mono-(C10), sesqui-(C15), di-(C20), sester-(C25), tri-(C30), tetra-(C40), and polyterpenes are some examples of these (Cn). In different cellular compartments, these natural compounds are biosynthesized via two different mechanisms. The plastid contains the 1-deoxyxylulose-5-phosphate (*DXP*) pathway, which is the source of mono-, di-, and tetra-terpenes [15,16]. The cytosol is the site of the mevalonate (MVA) pathway, which generates sesqui-, tri-, and poly-terpenes [16,17].

These substrates are transformed into the wide range of terpenoids that are present in plants by specialized enzymes called terpene synthases/cyclases. To improve the variety of plant terpenes, many terpenoids may be further altered by enzymatic and non-enzymatic methods [16,18]. Nearly half of all identified monoterpene and sesquiterpene synthases produce several products from a single substrate [19]. The product specificity of this enzyme class is very complex and cannot be deduced from the amino acid sequences. Many plant species, including the terpinene synthases of thyme and oregano [20,21,22,23], the linalool/nerolidol synthase from *Plectranthus amboinicus* [24], the linalool synthase, and terpinene synthase from *C. sativum*, have been cloned and reported [25]. The acyclic monoterpene alcohol linalool (3,7-dimethyl-1,6-octadien-3-ol) is produced by the enzyme linalool synthase (*LIS*) [26,27], which is the source of the floral smells found in many different plants, flowers, and spices. Due to its flavorful and aromatic qualities, it is widely used in processed foods, drinks, fragrances, cosmetics, waxes, soaps, and household detergents [28] (Figure 1).

The *TPS* and *LIS* genes of numerous plants, including *Citrus sinensis* [29], *Lathyrus odoratus* [30], *Dendrobium officinale* [31], and *Camellia sinensis*, have been discovered and functionally described [32]. Recent years have seen a plethora of research examining the impact of biological and non-biological inducers of the production of secondary metabolites, such as ultraviolet (UV) irradiation [33], trans-cinnamic acid [34], salicylic acid (SA), and methyl jasmonate (MeJA) [35,36,37,38,39,40,41,42]. It was shown that jasmonic acid (JA) and methyl jasmonate (its methyl ester) participate in signal transmission in plants, and they may regulate defense genes. Additionally, they are often used exogenously in plants to promote the production of secondary metabolites [35,39,40,41].

Despite the availability of various studies exploring the impact of elicitors such as MeJA on phenolic acid synthesis [43,44], we are not aware of any studies on linalool and the pattern of gene expression of linalool-related genes in Iranian coriander genotypes. The goal of this research was to investigate how MeJA affects the expression of key linalool synthase (*CsLINS*) and γ-terpinene synthase (*CsγTRPS*) genes, as well as the phenolic compound accumulation and linalool content in two Iranian coriander genotypes (Mashhad and Zanjan). Therefore, when the plant was stimulated with MeJA, we aimed to measure the resulting quantities of linalool, an essential chemical present in coriander leaves. We also made an effort to determine a link between the expression pattern of certain genes involved in the linalool production pathway and the altered levels of linalool in MeJA-treated leaves during the vegetative growth stage.

## 2. Results and Discussion

### 2.1. Changes in Linalool under MeJA Concentrations

We evaluated how different MeJA concentrations affected the primary coriander component linalool (Figure 2). The linalool content of both treated coriander ecotypes increased considerably after 24 h under MeJA treatment, as shown in Figure 2. Generally, the accumulation of linalool was greater in the Mashhad ecotype than in the Zanjan ecotype at all MeJA concentrations.

When the Mashhad ecotype was exposed to 10, 100, 150, and 200 µM MeJA, the level of linalool increased and was 1.10, 1.38, 1.48, and 1.21 times greater than that in the untreated plants. When these MeJA treatments were applied, the Zanjan ecotype produced 1.13, 1.24, 1.69, and 1.44 times more linalool than its control (Figure 2). Therefore, the two coriander ecotypes’ differing linalool amounts show that the quantity and mechanism of linalool production are linked to genotype-specific responses to abiotic stresses. In other words, our results suggest that 150 µM MeJA enhances the quantity of linalool present in various ecotypes of coriander, and that these elicitors have an impact on the expression patterns of genes involved in linalool production. Moreover, our results are consistent with those of other researchers [23,41,42] and imply that linalool levels in coriander plants vary according to genotype and dosage. Some research showed that the high levels of linalool seen following treatments with 100–150 µM MeJA represented the ideal dosage for treating various plant families [38,39,42]. According to these findings, caper plants exposed to 150 µM MeJA were able to produce flavonoids [40]. Linalool and other bioactive substances may be increased in a variety of plants as a result of MeJA’s induction of *LINS* and *TRPS* enzyme activities, according to certain studies [23,43]. The present study’s data suggest that MeJA may have had an impact on the rise in rutin quantity under 100 to 150 M treatment [39,41,42]. Our findings show that the administered MeJA had a discernible impact on both ecotypes metabolic traits. Last but not least, the increase in linalool accumulation brought on by MeJA stimuli may be associated with the activation of related genes and biosynthetic pathways that promote radical scavenging via phenolic components. Both ecotypes contained more linalool after MeJA treatment, although the Mashhad ecotype achieved the increase more quickly. This finding highlights the notion that linalool is known as the dominant component of the essential oil, rather than other terpenoids in diffident coriander genotypes.

This hypothesis is supported by the evidence that MeJA is the most potent inducer of the synthesis of terpinene and rosmarinic acid in *Thymus migricus* and *Agastache rugosa Kuntze* [23,45].

### 2.2. Total Flavonoid Content and Total Phenol Content (TPC) as a Function of MeJA Concentration (TFC)

There is little doubt that all MeJA concentrations impact the total flavonoid content (TFC) and total phenol content (TPC) levels of Iranian coriander leaves (Figure 3). The patterns of TPC and TFC alterations under the MeJA treatment in Mashhad and Zanjan coriander ecotypes likely mirrored those of linalool content changes (Figure 2 and Figure 3A,B). The accumulation of phenol and flavonoids was likewise altered by MeJA, although to a different degree, as they were greater in the Mashhad ecotype treated with MeJA than in the Zanjan ecotype. Recent studies have shown that MeJA, in its role as a signaling molecule, increases the accumulation of secondary metabolites in a variety of plant species [46,47]. Our findings are consistent with these studies and support their findings. It is probable that the enhanced expression of these genes in response to MeJA is a consequence of this increase in phenolic compounds.

The TPC quantity in the Mashhad ecotype was increased by 10, 100, 150, and 200 µM MeJA concentrations, which were 1.14, 1.32, 1.63, and 1.48 times greater than that in the untreated plants (Figure 3A). Accordingly, TPC values were determined for the Zanjan ecotype when exposed to the various concentrations of MeJA, and they were, respectively, 1.31-, 1.74-, 2.57-, and 1.04-fold higher than in the control (Figure 3A). Additionally, Mashhad coriander ecotypes treated with various MeJA concentrations resulted in TFC up to 7.68, 12, 22.52, and 15.71 mg QUE/g DW, with levels that were 1.61, 2.52, 4.74, and 3.28 times greater than that of the control plants (Figure 3B). Moreover, MeJA treatments boosted the TFC levels in the Zanjan ecotype, which were around 1.70, 2.37, 3.73, and 2.66 times higher than the levels in untreated plants (Figure 3B). These findings support the findings of other studies [23,41] and demonstrate that TPC and TFC variations in coriander leaves depend on ecotype. Additionally, in all ecotypes under consideration, TPC values in both untreated and all treated leaves were greater than the TFC values, which is consistent with previous results [23,41,42]. Our findings are in line with recent studies [48,49,50] which found that MeJA stimuli significantly affected TPC in various plants. MeJA has the capacity to boost the secondary metabolites accumulation in several plants, according to the same research [23,42,51].

The increased breakdown of bigger phenolic compounds into smaller ones, according to the hypothesis of Jaafar et al. [52], might lead to an increase in polyphenolic compounds. Thus, the gene expression involved in the linalool biosynthetic pathway was consistent with the increase in linalool and phenolic compound synthesis brought about by MeJA treatments. Linalool concentration in several plants has been shown to correlate with the expression of the linalool biosynthesis genes (*CsLINS* and *CsTRPS*) [29,30,31,32,43]. These results, which are consistent with other research [53], indicate that the genotype and dosage affect the amounts of TFC and TPC in thyme plants. Thymus species might be considered as possible sources of natural antioxidants, in addition to being rich sources of flavonoids and phenolics, according the aforementioned study.

By topically administering MeJA, some researches have tried to raise the total amount of phenolics or flavonoids in different plant species in order to boost antioxidant, antiadipogenic, and anti-proliferative activity [54].

Additionally, TPC values for both MeJA treated plants and controls were greater than TFC values, which is in line with earlier findings [54] for all thyme species.

### 2.3. MeJA Effects on the Expression of the CsγTRPS and CsLINS Genes

In this research, the real-time quantitative reverse transcription PCR (qRT-PCR) method was used to examine changes in the transcript levels of genes in the linalool pathway and the relationship between gene expression and linalool accumulation in Iranian coriander ecotypes exposed to various MeJA treatments (Figure 4). These findings unequivocally demonstrate that MeJA treatments considerably affected the mRNA transcript levels of mRNA in coriander. It is important to note that Mashhad ecotypes had higher transcription levels in the linalool biosynthesis pathway than did the Zanjan ecotypes, which may help to explain why MeJA produced more linalool, particularly in the final stages of linalool biosynthesis. To be more specific, as compared to untreated plants, the expression level of *CsγTRPS* in the Mashhad coriander ecotype increased from 6.28-fold at 10 µM MeJA to 9.69-fold at 150 µM MeJA and remained almost the same at 8.53-fold at 200 µM MeJA (Figure 4A). *CsLINS* expression increased significantly to 8.74-fold at 100 µM MeJA, quickly peaked at 17.84-fold with 150 µM MeJA, and then fell to the previous concentration (Figure 4B).

Treatment with MeJA 10 M significantly enhanced the expression level of *CsγTRPS* in the plants in the Zanjan ecotype as compared to the control plants (4.23-fold), while MeJA 10 and 100 M treatments showed no appreciable impact. At MeJA 150 M, *CsγTRPS* expression increased noticeably and was 6.56 times higher than in the control (Figure 4A). Following a 24 h period, MeJA treatment increased the Zanjan ecotype’s *CsLINS* gene expression by 4.24-fold at MeJA 10 M, 5.37-fold at MeJA 100 M, 8.73-fold at MeJA 150 M, and 6.59-fold at MeJA 200 M in comparison to the control plants. At all MeJA doses, transcript levels for every gene increased, and the linalool biosynthetic pathway’s Cs*LINS* and *CsγTRPS* of both ecotypes displayed the same pattern of expression (Figure 4). The MeJA treatment increased the expression of *CsLINS* and *CsγTRPS*, and their expression patterns matched the pattern of linalool accumulation perfectly. Moreover, the current research investigated the spraying of coriander ecotypes with high MeJA concentrations (200 M), resulting in a reduction in gene expression in comparison with other concentrations. These findings concur with those of Kianersi et al. [39,40,41] and Abdollahi et al. [42], who demonstrated that exogenously administered MeJA at high concentrations inhibited expression. In order to determine the maximum yield of secondary metabolites and to clarify their biosynthetic pathway(s), it would be helpful to understand how external stimuli affect secondary metabolite synthesis [35,36,55].

*CsγTRPS* expression in the two ecotypes under study was variably upregulated by MeJA treatments, and it showed the same pattern as linalool accumulation. In different organs of *Lycopersicon esculentum* cultivars and *Oenothera harringtonii*, the expression levels of genes relevant to linalool production showed similar results [43,44]. Our research showed that *CsLINS* is essential for the production of linalool. The rate of transcript levels and gene induction may change, depending on the stress and the type of plant.

According to earlier studies [43,44], MeJA, a vital enzyme in the linalool pathway, boosts linalool synthase (*LIS)* activity. Similarly, MeJA-treated *L. esculentum* and other species showed a considerable increase in the amount of gene transcription [43]. Despite the fact that our results clearly demonstrate the significance of *CsγTRPS* and *CsLINS* expression in linalool production in both ecotypes, the disparity in expression and linalool accumulation amounts suggests they are most likely related to coriander ecotypes.

A metabolic connection between the two compounds is suggested by the increase in linalool percentage reported in the studied samples [56]. For more specific information, Crocoll et al. [57] proposed that p-cymene is a byproduct of the premature release of the substrate from the active site, and that the synthesis of thymol and carvacrol takes place directly from the c-terpinene substrate. Additionally, they stated that p-cymene is a byproduct of the premature release of the substrate from the active site. The enhanced expression of these genes may be induced by an increase in the phenolic monoterpenes that these genes are known to produce. The absence or presence of ants near *Origanum vulgare* L. has been reported to induce an enhancement in the production and accumulation of phenolic monoterpenes, which is related to ants [58]. OvTPS2, CYP71D179/182, CYP71D180, and CYP71D178, among other genes involved in terpene biosynthesis, were found to be expressed more frequently when a Myrmica ant parasitized Origanum vulgare [58].

Our research demonstrated the close connection between *CsLINS* expression and linalool production. The expression pattern of *CsLINS* correlated with the increase in linalool in both Iranian coriander ecotypes after MeJA treatment (Figure 2 and Figure 4). For instance, in the Mashhad and Zanjan ecotypes, the expression of *CsLINS* at 150 µM MeJA after 24 h was 17.84 and 8.73 times greater, respectively, than in the control plants. In addition, at this concentration, the linalool concentrations were 1.48 times higher in the Mashhad ecotype and 1.68 times higher in the Zanjan ecotype than in untreated plants. Decreased production of linalool, flavonoid, and phenol levels at 200 µM MeJA compared to 150 µM MeJA were correlated with a lower *CsLINS* in the ecotypes under study. Similar patterns of *CsLINS* expression and linalool content may explain the critical role of these genes in regulating linalool production.

According to several studies [39,40,41,42,59], MeJA influences the transcript levels of genes, which is involved in the secondary metabolite biosynthesis pathways and leads to the accumulation of bioactive chemicals in a variety of plant species. According to Farooq et al. [60], exogenous treatment of MeJA further changed the activity of phenylalanine ammonia lyase (*PAL)*, polyphenol oxidases (*PPO)*, and carbamoyl-phosphate synthetase (*CAD)*, as well as their relative mRNA levels.

Rather than 200 µM MJ, 150 µM showed the greatest effects on the expression of *CsγTRPS* and *CsLINS* in the experimental setup. This suggests that the concentration of linalool increases along with the concentration of MeJA in a sample. Last but not least, our research shows that MeJA treatments significantly affected the levels of linalool, total flavonoid content (TFC), total phenolic content (TPC), and important genes involved in linalool production (*CsγTRPS* and *CsLINS*) in Iranian coriander leaves.

## 3. Materials and Methods

### 3.1. Plant Material and Growth Conditions

The seeds of the two ecotypes (Mashhad and Zanjan) of Iranian coriander (*C. sativum* L.) were sown in pots with a perlite–compost combination, and they were subsequently cultivated in a greenhouse with regulated lighting (16 h day/8 h night, with a photosynthetic photon flux density of 320 mol m^−2^ s^−1^) and temperature conditions (25/19 °C day/night).

### 3.2. MeJA Treatments

At 90 days old and in the vegetative development stage (i.e., only possessed root, stem, and leaf components), the potted *C. sativum* L. plants used in this study were treated with 10, 100, 150, and 200 µM MeJA, or distilled water, in the case of the control. Using a filter membrane with a 0.22 m MILLIPORE pore size, the MeJA solutions (SIGMA-ALDRICH) were fully sterilized. The final concentrations of MeJA solutions (10, 100, 150, and 200 µM) and distilled water (control) were subsequently sprayed onto the aerial parts of three coriander plants until runoff (1000 mL/treatment). For each treatment, three plants from each replication were taken into account (MeJA and distilled water). After the first 24 h of treatment, the treated and untreated uniform young leaves were collected and frozen in liquid nitrogen, and then stored at −80 °C until molecular and phytochemical analysis.

### 3.3. RNA Extraction, Complementary DNA (cDNA) Synthesis, and q-PCR Evaluation

According the manufacturer’s recommendations (SinaClon Bioscience Co., Karaj, Iran), the total RNA of coriander ecotypes’ uniform young leaves (100 mg frozen leaves) was extracted, and complementary DNA (cDNA) was produced using the kit’s two stages, in accordance with the manufacturer’s procedure [39], using the *β-actin* gene (a housekeeping gene) and gene-specific primers, as previously published [61]. The effects of different MeJA concentrations on the mRNA transcript levels of *CsγTRPS* and *CsLINS* were examined using the fold-change (2^−ΔΔCt^) technique, as previously reported [62] (Table 1). Additionally, three biological and technical duplicates were employed to analyze gene expression.

### 3.4. Determination of Linalool

The GC analysis was performed using a Hewlett-Packard 6890 gas chromatograph (Palo Alto, CA, USA) equipped with an FID and an electronic pressure control injector. Both a polar HP Innowax (polyethylene glycol) column and an apolar HP-5 column (both from HP; 30 m, 0.25 mm, 0.25 m film thickness) were used. The speed of the N2 carrier gas was 1.6 mL/min. The split ratio was 60:1. The analysis was conducted with the help of the following temperature program: the oven temperature was kept isothermally at 35 °C for 10 min, elevated from 35 to 205 °C at a rate of 3 °C/min, and then kept at 205 °C for 10 min. The injector and detector were maintained at temperatures of 250 and 300 °C, respectively. One liter of plain oil was injected into the samples. GC-MS analysis was carried out on a gas chromatograph HP 5890 (II) connected to a mass spectrometer with electron impact ionization (70 eV). The capillary column was an HP-5MS from Hewlett-Packard (30 m × 0.25 mm, 0.25 m film thickness). The samples were injected with 1 µL concentrated standard. The temperature of the column was set to increase by 5 °C per min from 50 °C to 240 °C. Helium served as the carrier gas, flowing at 1.2 mL/min with a 60:1 split ratio. The mass range and scan duration were 40–300 *m*/*z* and 1 s, respectively (Appendix A). Three replications were used in each injection.

The parameters mentioned in the literature [63] verified linalool compound. Finally, of all the essential oils, only the amount of linalool was reported in this work, as it is the main essential oil compound in coriander.

### 3.5. Flavonoid Contents and Total Phenolic Assay

Methanolic extracts of their total phenolic content (TPC) were obtained by shaking 1 gr of dried and crushed coriander leaf in 80% methanol for 24 h at room temperature (150 rpm). The TPC was measured after the extracts were filtered through three Whatman sheets [41]. First, 2.5 mL of diluted Folin-Ciocalteu reagent, 2 mL of sodium carbonate, and 0.5 mL of each sample’s methanolic extract (7.5 percent) were combined. After 15 min at 45 °C, 765 nm absorbance was recorded. The tannic acid equivalent in mg/g dry weight (DW) was used to determine TPC (Appendix A).

Total flavonoid content (TFC) was determined using aluminum chloride colorimetry [64]. First, 0.25 mg of each sample extract was mixed with 1.25 mL of water and 0.75 mL of sodium nitrate. After 300 s of dark incubation, 0.15 mL of 10% aluminum chloride was applied. Each sample received 0.275 mL water and 0.5 mL sodium hydroxide solution. The reaction solution’s 510 nm adsorption and TFC were read (Appendix A).

### 3.6. Statistical Evaluation

Utilizing factorial experiments with a totally randomized design, the linalool content of two Iranian ecotypes of coriander treated with varied MeJA doses was examined for TFC, TPC, and gene expression (CRD). In every experiment, there were three replications. The statistical program SPSS 16 was used to run an ANOVA on all of the data. To compare the means, the Duncan’s multiple range test was used (DMRT).

## 4. Conclusions

We demonstrated that the application of exogenous MeJA in coriander, particularly Iranian coriander ecotypes, may increase the levels of linalool, TFC, and TPC, as well as the transcript levels of crucial genes which are involved in the linalool pathway, such as *CsγTRPS* and *CsLINS*. Further study is required to better understand how the expression of additional genes connected to this pathway interact with the buildup of phenolic compounds in response to MeJA and other treatments which involve abiotic stressors. Future research could make it possible to genetically alter this process to increase the synthesis of beneficial chemicals in coriander.

## Figures and Tables

**Figure 1 genes-13-01717-f001:**
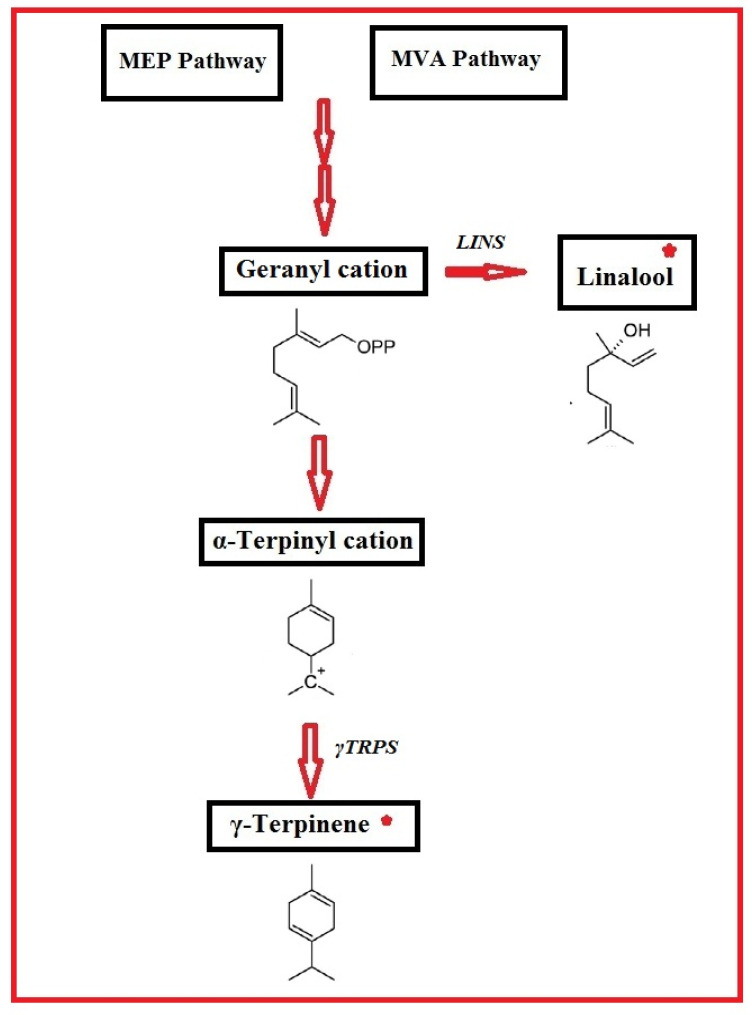
Conversion of geranyl diphosphate to (S)-linalool by *LINS*, and to other monoterpene products by *TRPS*.

**Figure 2 genes-13-01717-f002:**
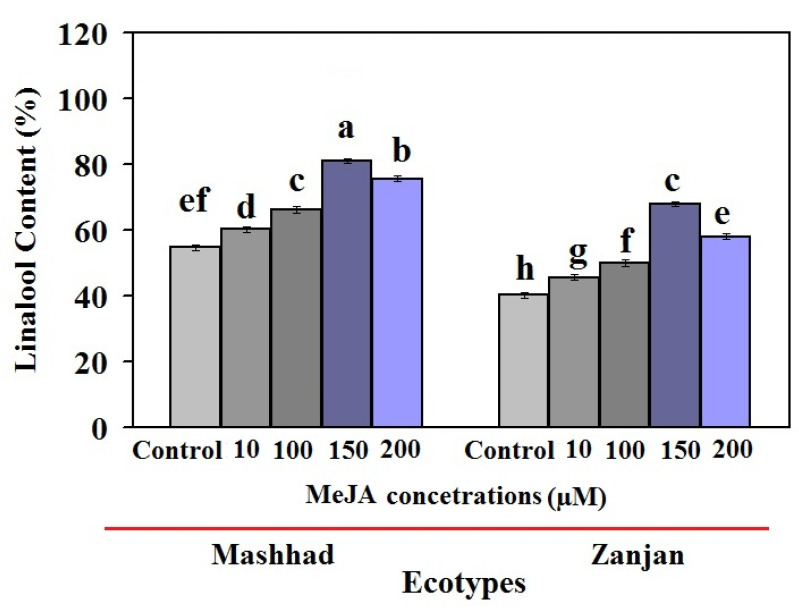
Linalool content of Iranian coriander ecotype leaves after various MeJA treatments. According to Duncan’s test, bars with different lettering indicate significance at a 1% level of probability. Error bar are shown in percent of standard deviation.

**Figure 3 genes-13-01717-f003:**
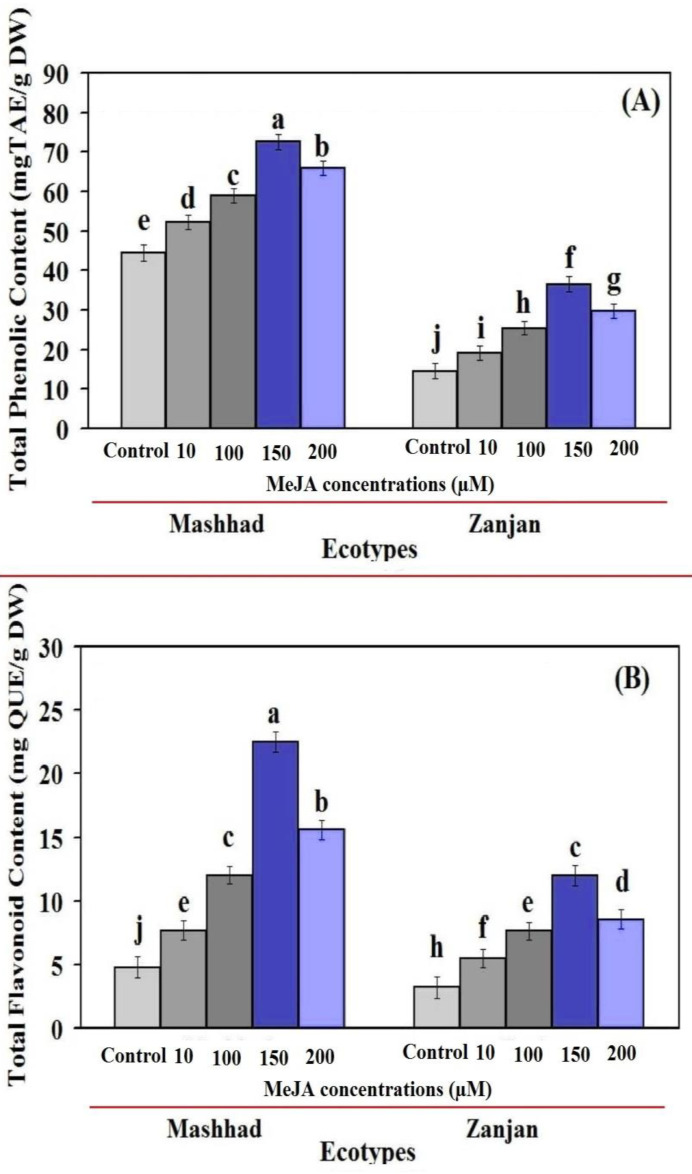
The impact of various MeJA concentrations on the TPC (**A**) and TFC (**B**) of leaf extracts from two Iranian coriander ecotypes. The data are shown as mean standard deviations (n = 3). According to Duncan’s test, distinct letters in each column indicate significance at a 1% level of probability. Error bars are shown in percent of standard deviation.

**Figure 4 genes-13-01717-f004:**
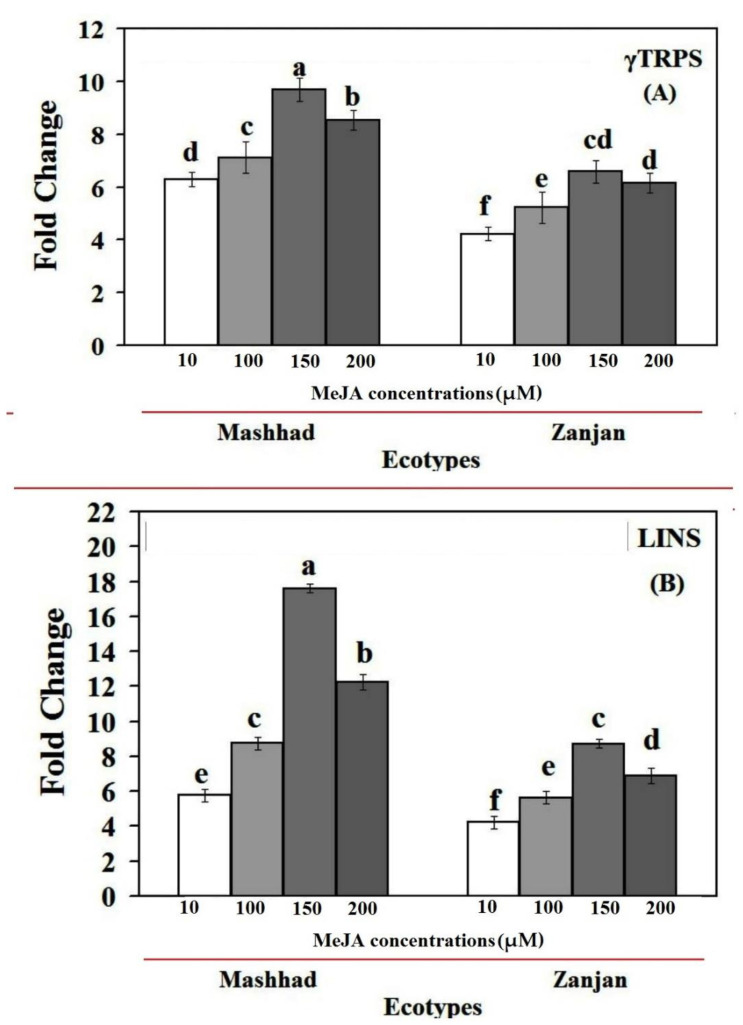
Linalool biosynthetic pathway genes *CsγTRPS* (**A**) and *CsLINS* (**B**) are expressed in the leaves of control (untreated) and MeJA-treated coriander plants, respectively. The Ct values formed the basis of qRT-PCR. The reference gene actin was used to standardize the Ct value for each sample. According to Duncan’s test, bars with different letters are substantially (*p* ≤ 0.01) different. Standard error values are shown by error bars.

**Table 1 genes-13-01717-t001:** Primers used for qRT-PCR analysis.

Real-Time Primers	Sequences (5′ to 3′)
*CsγTRPS F* *CsγTRPS R*	CGAAATGGTGGAAGGACACAGA GTAATAGCAGCGAGCACCTT
*CsLINS F* *CsLINS R*	GAGAAGGACTTGCATGCTACTG GACATCTGCACGGATACCT
*β-Actin* *β-Actin*	GACGAGGATGAGGCAGAGTT GGAGCATCAGAAACAGAGG

## Data Availability

All data is contained within this article.

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
