# Peer review of "Methyl Jasmonate Induces Genes Involved in Linalool Accumulation and Increases the Content of Phenolics in Two Iranian Coriander (Coriandrum sativum L.) Ecotypes"

_genes, 2022, doi:10.3390/genes13101717_

Round 1

Reviewer 1 Report

The authors investigate the effect of MeJA on the transcript accumulation of Iranian coriander. MeJA applied in high micromolar concentrations increases the level of coriander and lead to an increase in terpenoids and phenolics as well as increase in some transcripts related to terpenoid formation. The story is in line with many other publications on the effects of MeJA on plant natural product biosynthesis. The data seem only partly o.k.. In addition, presentation and overall outlay of  the manuscript requires substantial revision.

1.  The authors only present Meta-data. Please show original chromatograms and also standard curves for photometric measurements of phenolics and tannins. TPC and TFC. It is unfortunate, that not even TLC plates are shown, no HPLC-data, so there is no exact quantification of these compounds possible, the methodology is more than 50 years old and not up to date. Result is a low impact.

2.  qRT-PCR transcript data and individual Figures and legends should be better presented. These letters on individual bars mean nothing to me, and are very confusing. Figure legends are also not appropriate and should be more specific explaining in detail the meta-data. What does the content of 100 % linalool relate to 100 % of what? In addition, the reader, it is not clear how the leaves were chosen, old, young, of similar age? Please be specific.

3.      Only a single time point was chosen for analysis, to my opinion at least two, better three times points are required to judge the relevance to the metabolite data. Why 24 hours for metabolite analysis? How do the authors know that this is the best time points? Time points for transcript accumulation was identical? Earlier or later, based on which previous data?

4.      The overall introduction, results and discussion part should be better structured, and less redundant. E.g. Lines 91-102 are partly redundant.

5.      Line 108 add reference and remove statements like “for the first time”!

6.  Add structures to Figure 1 and modify genaryl to geranyl, this is kind of embarrassing that none of the co-authors saw this, did they read this MS?

7.     Line 130: why dramatic?? This is a slight increase at a single time point, improve! In addition, I do not see any fluctuation, be careful not to use inappropriate wording.

8.      Line 132-133 should be compared to line 147-148. Just one example of redundancy. To me it shows that this manuscript was written and read in a sloppy way. Another example, line 283-286, remove!

9.      Line 151: what kind of metabolites?

10. Figure 4: Fold-change of what, the controls should be rated as 1 and the fold change of individual TRPS and LINS are 2- and 3-fold respectively, is this correct? 6-fold more than actin, what sense does this make, if you do not know if actin transcript is affected?

11.  Determination of linalool: please show the chromatogram in a separate Figure, show the differences. Also include the original data of the photometric measurements as a supplemental File.

Finally Data Availibility: The primary data are not contained in this article, please provide the relevant ones in supplemental files and include the GC-chromatograms +/- MeJA, appropriately scaled as an extra Figure.

Author Response

Dear reviewer,

Thank you for your consideration and valuable points.

We have provided the proper answer to each query point to point and addressed in manuscript using yellow highlighted parts, as well as we added the supplementary file. 

Best Regards

The authors investigate the effect of MeJA on the transcript accumulation of Iranian coriander. MeJA applied in high micromolar concentrations increases the level of coriander and lead to an increase in terpenoids and phenolics as well as increase in some transcripts related to terpenoid formation. The story is in line with many other publications on the effects of MeJA on plant natural product biosynthesis. The data seem only partly o.k.. In addition, presentation and overall outlay of the manuscript requires substantial revision.

 Thank you for your positive opinion and interesting.

  1. The authors only present Meta-data. Please show original chromatograms and also standard curves for photometric measurements of phenolics and tannins. TPC and TFC. It is unfortunate, that not even TLC plates are shown, no HPLC-data, so there is no exact quantification of these compounds possible, the methodology is more than 50 years old and not up to date. Result is a low impact.

Response: Thank you for your consideration and point. The standard curve of total phenolic content and total flavonoid content were added to supplementary file.

As you know there is different and important reposts and manuscripts about the measuring method of TPC and TFC (Salami et al., 2016; Kianersi et al. 2021, 2022), in different plants. So, we selection criteria was other studies which previously showed that the measuring methods which have been used in some plants (Zhang et al., 2015; Salami et al, 2016; Kianersi et al. 2021 in Industrial Crops and Products, Food chemistry, international journal of molecular science).

Also, as these compounds, TPC and TFC, can be detected by spectrophotometrically, we used this method for measuring instead by TLC, and HPLC that is normal and functional method. 

Finally, with respect to the referee, we wrote and put necessary data and figures in manuscript as well as chromatograms related to the Linalool standard and Coriander linalool in supplementary files. On the other hand, we can not share our raw data and all figures with anyone, due to copyright law and roles.

  1. RT-PCR transcript data and individual Figures and legends should be better presented. These letters on individual bars mean nothing to me, and are very confusing. Figure legends are also not appropriate and should be more specific explaining in detail the meta-data.

Response: Thank you for your opinion. All figure of manuscript were replaced with new figure. However, with respect to the referee, as explained in statistical analysis and result and discussion sections, we thoroughly and accurately was done all statistical analyzes required to interpret the experiment data as well as we showed the kind of statistical design (factorial experiments based on the completely randomized design), number of replicate (Three replications) and the method of mean comparison (Duncan's multiple range tests). Also, the p-values with a short description in section of figure captions (figure legends) (Bars with different letters are significantly (P<0.01) different according to Duncan’s test.).

  1. What does the content of 100 % linalool relate to 100 % of what?

Response: The amount of volatile compounds were and are reported as percent (%) in different manuscripts (Salami et al., 2016; Tohidi et al., 2019, 2020; Kianersi et al., 2021). So, the sign "%" is indicator of linalool content in control and treatment plants in our manuscript.  

  1. In addition, the reader, it is not clear how the leaves were chosen, old, young, of similar age? Please be specific.

Response: Thank you for your opinion. As explained in material and method section and application of MeJA treatments section, the young leaves of Coriander plant were selected and harvested, when they were 90 days old and at the vegetative growth stage. On the other words, the three uniform young leaves coriander of plant including 3 replicates were harvested from treatment and non-treatment plants as random from different point of plants.

  1. Only a single time point was chosen for analysis, to my opinion at least two, better three times points are required to judge the relevance to the metabolite data. Why 24 hours for metabolite analysis? How do the authors know that this is the best time points? Time points for transcript accumulation was identical? Earlier or later, based on which previous data?

Response: Thank you for your point. Respecting the reviewer, there is different and important manuscripts about the application of elicitors (MeJa and SA) in various time points (Kianersi et al. 2020a,b, 2021, 2022) in different plants, which had previously shown that the treatment time point of 24 hours was the best time point and considered because at the beginning of testing and treatment, elicitors do not have an immediate effect on the plant and their results need time. So, we selection criteria was these studies.

  1. he overall introduction, results and discussion part should be better structured, and less redundant. E.g. Lines 91-102 are partly redundant.

Response: Thank you for your opinion. On the recommendation of the honorable reviewer, unnecessary sentences were removed in the text.

  1. Line 108 add reference and remove statements like “for the first time”!

Response: Thank you for your opinion. The reference was inserted in the text as well as unnecessary sentences were removed in the text.

  1. Add structures to Figure 1 and modify genaryl to geranyl, this is kind of embarrassing that none of the co-authors saw this, did they read this MS?

Response: Thank you for your opinion. On the recommendation of the honorable reviewer, corrections and new figure were inserted in the text.

  1. Line 130: why dramatic?? This is a slight increase at a single time point, improve! In addition, I do not see any fluctuation, be careful not to use inappropriate wording.

Response: Thank you for your opinion. Corrections were inserted in the text.

  1. Line 132-133 should be compared to line 147-148. Just one example of redundancy. To me it shows that this manuscript was written and read in a sloppy way. Another example, line 283-286, remove!

Response: Thank you for your opinion. Corrections were inserted in the text as well as unnecessary sentences were removed in the text.

  1. Line 151: what kind of metabolites?

Response: Corrections were inserted in the text.

  1. Figure 4: Fold-change of what, the controls should be rated as 1 and the fold change of individual TRPS and LINS are 2- and 3-fold respectively, is this correct? 6-fold more than actin, what sense does this make, if you do not know if actin transcript is affected?

Response: According to Schmittgen and Livak (2008) formula, we computed the real-time data using fold change methods that used in different papers (Sun et al, 2012; Fatemi et al, 2019, Kianersi et al., 2020,2021, 2022)

                                                 Fold change = 2−ΔΔC

By using the fold change methods to analyze relative real-time PCR data (Schmittgen and Livak 2008), the PCR efficiency of the target and internal control genes are calculated in the equation and differences in the efficiency between them will be accounted for in the calculation. This form of the equation may be used to compare the gene expression in two different samples. We used calling the untreated control and the calibrator. In this way the data interpreted that “the expression of the gene of interest relative to the internal control in the treated sample compared with the untreated control”. The amount of control is one. The change amount of mRNA transcript could be different based on the treatments.

  1. Determination of linalool: please show the chromatogram in a separate Figure, show the differences. Also include the original data of the photometric measurements as a supplemental File.

Response: On the recommendation of the honorable reviewer, chromatograms related to the Linalool standard and Coriander linalool were added to supplementary file. However, due to copyright law and roles. We can not share our original data and raw data, all figures with anyone.

  1. Finally Data Availability: The primary data are not contained in this article, please provide the relevant ones in supplemental files and include the GC-chromatograms +/- MeJA, appropriately

Response: With respect to the referee, we wrote and put all necessary data and figures in manuscript. On the other hand, we can not share our raw data and all figures with anyone, due to copyright law and roles. However, we added the list of identified metabolites in supplementary file.

Reviewer 2 Report

The current manuscript discusses the effect of MeJA on the metabolites and related gene expressions in coriander with multiple application values. The work is in time and interesting. Before further consideration, I have several concerns.

1. Multiple metabolites can be identified via GC-MS, the authors just showed the dynamic pattern of linalool content. How about other metabolites? List the identified metabolites as a supplementary table.

2. The leaf after 24 h treatment was collected for this analysis. What the reference?

3. There are serval key genes related to LIN, TPC, and TFC metabolism. Why the authors only analyzed the two genes, TRPS and LINS?

4. L124-127, the authors pointed the level of linalool in the MeJA treated plants was 2.54, 3.53, 5.21, and 4.74 times greater than in untreated plants. But the Figure 2 just showed the linalool content ranged from about 55% to 85%. How the authors calculate the results? The same puzzlement in other bar graph analyses, like L172-177, L222-225, L232-233.

Minor comments.

1. Gene names should be italic.

2. The use of abbreviation in the manuscript is irregular. Latin names should be abbreviated after the first occurrence.

3. L114, point out the position of LINS and TRPS in the pathway.

4. L148, there is no gene expression in this part.

5. L319, -80.

6. L346, please add the description of the sample pre-treatment for the GC-MS analysis. SPME?

Author Response

Thank you so much for your consideration and giving us an opportunity to revise the manuscript. Regarding to the written comments by your potential reviewers, we appreciate the points mentioned. We have provided the proper answer to each query point to point and addressed in the manuscript using green highlighted parts, as well as we added the supplementary file.

1-Multiple metabolites can be identified via GC-MS, the authors just showed the dynamic pattern of linalool content. How about other metabolites? List the identified metabolites as a supplementary table.

Response: With respect of honorable reviewer, as we explained in manuscript, the goal of this study was to investigate how MeJA affected the expression of key linalool synthase (CsLINS) and γ-terpinene synthase (CsγTRPS) genes, and the linalool content of two Iranian coriander genotypes (Mashhad and Zanjan). Therefore, we just reported the linalool compounds as main ingredient of coriander plant. On the other words, we made an effort to establish a link between the pattern of expression of certain genes involved in the linalool production pathway and the altered levels of linalool in the leaves of MeJA-treated Coriander plants during the vegetative growth stage.

However, we added the list of identified metabolites in supplementary file.

  1. he leaf after 24 h treatment was collected for this analysis. What the reference?

Response: Thank you for your point. Respecting the reviewer, there is different and important manuscripts about the application of elicitors (MeJa and SA) in various time points (Kianersi et al. 2020a,b, 2021, 2022) in different plants, which had previously shown that the treatment time point of 24 hours was the best time point and considered because at the beginning of testing and treatment, elicitors do not have an immediate effect on the plant and their results need time. So, we selection criteria was these studies.

  1. here are serval key genes related to LIN, TPC, and TFC metabolism. Why the authors only analyzed the two genes, TRPS and LINS?

Response: With respect of honorable reviewer, as we explained in manuscript, the objectives of this research were to ascertain the effects of methyl jasmonate (MeJA) on LIN content, total phenolic content (TPC), and total flavonoid content (TFC), as well as changes of the expression of genes involved in linalool biosynthetic pathway (CsγTRPS and CsLINS). On the other words, we made an effort to establish a link between the pattern of expression of certain genes involved in the linalool production pathway and the altered levels of linalool in the leaves of MeJA-treated Coriander plants during the vegetative growth stage.

4- L124-127, the authors pointed the level of linalool in the MeJA treated plants was 2.54, 3.53, 5.21, and 4.74 times greater than in untreated plants. But the Figure 2 just showed the linalool content ranged from about 55% to 85%. How the authors calculate the results? The same puzzlement in other bar graph analyses, like L172-177, L222-225, L232-233.

Thank you for your opinion. Corrections were inserted in the text.

Minor comments.

  1. ene names should be italic.

Response: Thank you for your opinion. Corrections were inserted in the text.

  1. The use of abbreviation in the manuscript is irregular. Latin names should be abbreviated after the first occurrence.

Response: Thank you for your opinion. Corrections were inserted in the text.

  1. 114, point out the position of LINS and TRPS in the pathway.

Response: Thank you for your opinion. Corrections were inserted in the text.

  1. L148, there is no gene expression in this part.

Response: Corrections were inserted in the text.

  1. 319, -80.

Response: Corrections were inserted in the text.

  1. L346, please add the description of the sample pre-treatment for the GC-MS analysis. SPME?

Response: Thank you for your opinion. As explained in material and method section for application of MeJA treatments, the potted Coriandrum sativum L. plants in this research were treated with 10, 100, 150, and 200 µM MeJA as well as distilled water (as a control) when they were 90 days old and at the vegetative growth stage (i.e., only possessed root, stem, and leaf components). MeJA (SIGMA-ALDRICH) solutions were thoroughly sterilized using a filter membrane that had a 0.22 m MILLIPORE pore size. The aerial sections of three coriander plants were then sprayed with final concentrations of MeJA solutions (10, 100, 150, and 200 µM) and distilled water (control) till runoff (1000 mL/treatment). Three plants per replication were taken into consideration and gathered for each treatment (MeJA and distilled water). The treated and untreated uniform young leaves were collected after the first 24 hours of treatment, frozen in liquid nitrogen, and stored at -80°C for molecular and phytochemical analyses. Also, we measured the linalool compound by GC and GC-MS method based on previously report (Msaada et al., 2009).

Round 2

Reviewer 1 Report

The authors have addressed all comments of the reviewer, but not all of these responses are of sufficient quality.

1.    The title is somewhat misleading. It only addresses the terpenoid part of the report, while phenolics and flavonoids are not mentioned. The title should be modified. What is “Phytochemical accumulation?” Suggestion: Methyl jasmonate induces genes involved in linalool accumulation and increases the content of phenolics in two Iranian Coriander (Coriandrum sativum L.) ecotypes

2.    Asking for raw-data and chromatograms: in S1 the authors display two chromatograms, a linalool standard and the corresponding peak/chromatogram from a control plant. Please mark the linalool peak in each chromatogram (including m/z) and add C (after 150 µM treatment) a chromatogram scaled to the same mVolt as the control for comparison. Clearly mark the peak and add the total area or any other quantification detail into the legend. In addition, the comment that the raw data cannot be released is inappropriate. These raw data are usually required in all higher impact journals, should be deposited in a data repository, be made available to the public several years to reduce experimental fraud and ensure scientific quality. Statements like this from the corresponding author could raise doubts on data integrity.

3.    Still many minor (syntax) errors:

Line 45: ...has been used since antiquity as a spice, as an odorant, in folk medicine…

Line 96: affects

Line 190: Starting with: Furthermore… delete this sentence (last one on page 6) completely…what kind of solvents should this be??

Line 270: pathways

Line 275: delete surprisingly….´Rather than 200 µM MJ, 150µM showed the largest effects on the expression of….in the experimental setup.

Line 330 5 °C…from 50 °C to 240 °C...

Line 333: verifies what exactly?

Line 362: remove … for the first time…

Line 368/369: how exactly should this be done? Can coriander be engineered by CRISPR/CAS?

Author Response

Dear reviewer,

Thank you so much for your consideration and for giving us an opportunity to revise the manuscript.

We have provided the proper answer to each query point to point and addressed here using yellow highlighted parts in the text, too as well as we added the supplementary file. 

Review 1

The authors have addressed all comments of the reviewer, but not all of these responses are of sufficient quality.

Thank you for your positive opinion and interest.

The title is somewhat misleading. It only addresses the terpenoid part of the report, while phenolics and flavonoids are not mentioned. The title should be modified. What is “Phytochemical accumulation?” Suggestion: Methyl jasmonate induces genes involved in linalool accumulation and increases the content of phenolics in two Iranian Coriander (Coriandrum sativum L.)

Response: The new subject was added to the text.

  1. Asking for raw-data and chromatograms: in S1 the authors display two chromatograms, a linalool standard and the corresponding peak/chromatogram from a control plant. Please mark the linalool peak in each chromatogram (including m/z) and add C (after 150 µM treatment) a chromatogram scaled to the same mVolt as the control for comparison. Clearly mark the peak and add the total area or any other quantification detail into the legend. In addition, the comment that the raw data cannot be released is inappropriate. These raw data are usually required in all higher impact journals, should be deposited in a data repository, be made available to the public several years to reduce experimental fraud and ensure scientific quality. Statements like this from the corresponding author could raise doubts on data integrity.

Response: Thank you for your opinion. On the recommendation of the honorable reviewer, corrections and new figure (c) chromatograms related to the 150 µM MeJA treated-Coriander plant were inserted to supplementary file

  1. Still many minor (syntax) errors:

Line 45: ...has been used since antiquity as a spice, as an odorant, in folk medicine…

Response: Thank you for your opinion. Corrections were inserted in the text.

Line 96: affects

Response: Thank you for your opinion. Corrections were inserted in the text.

Line 190: Starting with: Furthermore… delete this sentence (last one on page 6) completely…what kind of solvents should this be??

Response: Thank you for your opinion. Corrections were inserted in the text as well as unnecessary sentences were removed in the text. The mean of solvents was elicitors.

Line 270: pathways

Response: Thank you for your opinion. Corrections were inserted in the text.

Line 275: delete surprisingly….´Rather than 200 µM MJ, 150µM showed the largest effects on the expression of….in the experimental setup.

Response: Thank you for your opinion. Corrections were inserted in the text as well as unnecessary sentences were removed in the text.

Line 330 5 °C…from 50 °C to 240 °C...

Response: Thank you for your opinion. Corrections were inserted in the text.

Line 333: verifies what exactly?

Response: Thank you for your opinion. Corrections were inserted in the text. The mean was linalool compound.

Line 362: remove … for the first time…

Response: Thank you for your opinion. Corrections were inserted in the text as well as unnecessary words were removed in the text.

Line 368/369: how exactly should this be done? Can coriander be engineered by CRISPR/CAS?

Response: Thank you for your point. Our proposal is to create a transgenic cilantro plant to produce more secondary metabolites using biotechnology techniques. It is also possible to use the CRISPR/CAS technique as a new genetic system to create desired characteristics in plants such as coriander. However, this technique has advantages and disadvantages that should be thoroughly investigated (Das et al., 2022: CRISPR/Cas Genome Editing in Engineering Plant Secondary Metabolites of Therapeutic Benefits).

Best regards,

Reviewer 2 Report

A few minor changes are needed.

L105-106. The gene names should be presented as LINS and TRPS, not CsLINS or CsγTRPS.

L116. How did the authors obtain the 1.38 times with 10 µM MeJA treatment? Please check all.

L318. The suggestion I made last time was to add the description of the pre-treatment for the GC-MS analysis after sample collecting. How did the authors collect the aroma components for linalool determination? Through solid-phase microextraction or others.

L326. 250°C and 300°C? Please check L330-331.

Author Response

Dear reviewer,

Thank you so much for your consideration and for giving us an opportunity to revise the manuscript. 

We have provided the proper answer to each query point to point and addressed here using green highlighted parts in the text.

A few minor changes are needed.

L105-106. The gene names should be presented as LINS and TRPS, not CsLINS or CsγTRPS.

Response: Thank you for your opinion. Corrections were inserted in the text.

L116. How did the authors obtain the 1.38 times with 10 µM MeJA treatment? Please check all.

Response: Thank you for your point. With respect to the referee, the amount of linalool were 54.68% and 75.62 % in control and 10 µM MeJA treated-plants, respectively. So, the ratio linalool is 1.38 times (75.62÷ 54.68) relative to control plants.

L318. The suggestion I made last time was to add the description of the pre-treatment for the GC-MS analysis after sample collecting. How did the authors collect the aroma components for linalool determination? Through solid-phase microextraction or others.

Response: Thank you for your opinion. Leaves of Coriander plants (100 g) were subjected to hydrodistillation for 90 min. The distillate was extracted with 2-methyl-butane (v/v) and dried over anhydrous sodium sulphate. The organic layer was then concentrated, at 30°C using a Vigreux column and the resulting essential oil was subsequently analyzed (Msaada  et al., 2006, 2009).

L326. 250°C and 300°C? Please check L330-331.

Response: With respect of honorable reviewer, as we explained in manuscript, the number of temperatures are correct. All methods for doing this part have done based on previously reports (Msaada et al., 2006, 2009) that is special for coriander plant.

 Sincerely,

Round 3

Reviewer 1 Report

Please clarify the following discrepancy:

Figure 2A shows a steady increase of linalool In Mashbad genotype. However, line 110, the authors state that the linanool content at 10 µM, 100 µM, 150 µM and 200 µM rises 1.38, 1.10, 1.48, 1,21 times. This means that  at 100 µM it is lower than at 10 µM? In addition, Figure S1 shows a rather strong increase, 6 fold in the GC-MS chromatogram of induced versus non-induced, not a 10-fold induction. so much more, why that? Please also state in the Figure S1 the concentration of the standard in the legend and that the same or different amounts of tissue was used.

Regarding additional terpenoids in Table S1: were these also induced similarly? Apparently, not since the percentage of linalool rose at the expense of other metabolites, is this correct. Add a comment in the manuscript on this in the manuscript.

Author Response

Dear Reviewer

Thank you so much for your consideration and giving us an opportunity to revise the manuscript. We have provided the proper answer to each query point to point and addressed here using yellow (first reviewer), highlighted parts in the text, too as well as we added the supplementary file.
Sincerely,

Review 1

Comments and Suggestions for Authors Please clarify the following discrepancy: Thank you for your positive opinion and interesting.

Figure 2A shows a steady increase of linalool In Mashbad genotype. However, line 110, the authors state that the linanool content at 10 μM, 100 μM, 150 μM and 200 μM rises 1.38, 1.10, 1.48, 1,21 times. This means that at 100 μM it is lower than at 10 μM?

Response: Thank you for your point. Corrections were inserted in the text.

In addition, Figure S1 shows a rather strong increase, 6 fold in the GC-MS chromatogram of induced versus non-induced, not a 10-fold induction. So much more, why that?

Response: Thank you for your opinion and point. With respect to the referee, the GC-MS chromatogram are indicate the area and tend of linalool and compounds that how they changed. So, these can not exactly show the differences and amounts. We calculated based on other parameters like sample weight, concentration of treatments and etc by formula. Finally, the changing amount is dependent on them.

Please also state in the Figure S1 the concentration of the standard in the legend and that the same or different amounts of tissue was used.

Response: Thank you for your consideration. Corrections were inserted in the supplementary file.

Regarding additional terpenoids in Table S1: were these also induced similarly? Apparently, not since the percentage of linalool rose at the expense of other metabolites, is this correct. Add a comment in the manuscript on this in the manuscript.

Response: Thank you for your opinion and point. New comments were inserted in the text. With respect to the referee, this finding highlights the notion that Linalool was known as the dominant component of the essential oil rather than other terpenoids in diffident coriander genotypes. On the other hand, the goal of this research was to investigate how MeJA affected the linalool as dominant component and expression of key genes involved in linalool biosynthetic pathway, synthase (CsLINS) and γ-terpinene synthase (CsγTRPS). So, we just selected and concentrated on linalool compound. Therefore, finally, of all the essential oils, only the amount of linalool was reported in this work, as it is the main essential oil compound in coriander.

Round 4

Reviewer 1 Report

The value of the peak in the GC chromatogram in Supplemental Figure 1A should correspond to a single (!) value. This may be based on a standard curve, but it corresponds to a single injection. Either show the standard curve for linalool or present a single number. How much does an area of 21789 correspond to in terms of µg, ng or pmol/fmol? The authors should know how many µl were injected into the GC-system and based on this, how many pmol or fmol the number 21789 corresponds to.  In addition, how exact is this measurement? Even if computational calculations indicate a five o even 6 digit number, the number of digits should be limited to a number that can be stated with confidence, usually a maximum of 2 or 3 digits like 1.25, 12.2, or 227 (these numbers are just arbitrary examples). Thank you!

One more comment is required in the results on the rise or fall of the remaining terpenoids. The GC-chromatograms should indicate if there is an overall rise or if a single compound, linalool, increases at the expense of others. Exact numbers are not required. 

Author Response

Dear Reviewer, 

Thank you so much for your consideration and giving us an opportunity to revise the manuscript. We have provided the proper answer to each query point to point and addressed here using yellow (first reviewer), highlighted parts in the text, too as well as we added the supplementary file.

Comments and Suggestions for Authors

Thank you for your valuable points. I sincerely appreciate your time to review this article. We tried to answer your valuable questions well.

The value of the peak in the GC chromatogram in Supplemental Figure 1A should correspond to a single (!) value. This may be based on a standard curve, but it corresponds to a single injection. Either show the standard curve for linalool or present a single number.

Response: Thank you for your point and opinion. The curve of figure 1A that wrote the linalool is related to linalool compound that was identified by set based on linalool standard from all compounds that there were in the plant.

How much does an area of 21789 correspond to in terms of µg, ng or pmol/fmol?

Response: Thank you for point. Whit respect to the reviewer, the number 21789 is just the area which GC-set indicated, after measuring the amount of linalool compound based on µg of linalool standard.

The authors should know how many µl were injected into the GC-system and based on this, how many pmol or fmol the number 21789 corresponds to. 

Response: Thank you for your consideration. As we explained in material and method section, 1 µl concentrated standard and samples were injected. So, when you have injected the samples, the linalool amount (%) will be calculated and reported based on the area of the obtained curve that it depends on powder and extract weight of standard and samples compound.

In addition, how exact is this measurement? Even if computational calculations indicate a five o even 6 digit number, the number of digits should be limited to a number that can be stated with confidence, usually a maximum of 2 or 3 digits like 1.25, 12.2, or 227 (these numbers are just arbitrary examples). Thank you!

Response: Thank you for your guidance and points. With respect of honorable reviewer, as we explained in manuscript, all methods for doing this part have done based on previously reports (Msaada et al., 2006, 2009) that is special for coriander plant. All data obtained directly from GC-set for analyzing. On the other hand, three replications were used in each injection and experiment as well as the chromatography peak of linalool was confirmed according to the retention time and FID system of GC. The number of digits got directly from GC-set from computer that can be including the 2 till different digits based on the compounds area amount.

One more comment is required in the results on the rise or fall of the remaining terpenoids. The GC-chromatograms should indicate if there is an overall rise or if a single compound, linalool, increases at the expense of others. Exact numbers are not required. 

Response: Thank you for your opinion and point. With respect to the referee, the goal of this research was to investigate how MeJA affected the linalool as dominant component and expression of key genes involved in linalool biosynthetic pathway, synthase (CsLINS) and γ-terpinene synthase (CsγTRPS). So, we just selected and concentrated on linalool compound, as it is the main essential oil compound in coriander as well as there is near to 15 terpenoids that we identified in extracts (table 1). As they are not dominant compound in coriander, so we can not explained that the how was the changing of them (increase or decrease) after treatments. On the other hand, with respect to the referee, as you know, additional experiments require more time and in present project we are not left with enough time for adding these experiments, and it is not possible at the moment because we do not access to plant material and Covid-19 situation. So unfortunately we would not be able to add these additional experiments in the current manuscript, but we will do additional experiments in our next planned projects.

I look forward to hearing your positive response.

Wish you all the best,

Sincerely